# Linearly Interpretable Concept Embedding Model for Text Classification

## Abstract

Despite their success, Large-Language Models (LLMs) still face criticism due to their lack of interpretability. Traditional post-hoc interpretation methods, based on attention and gradient-based analysis, offer limited insight as they only approximate the model's decision-making processes and have been proved to be unreliable. For this reason, Concept-Bottleneck Models (CBMs) have been lately proposed in the textual field to provide interpretable predictions based on human-understandable concepts. However, CBMs still face several criticisms for their architectural constraints limiting their expressivity, for the absence of task-interpretability when employing non-linear task predictors and for requiring extensive annotations that are impractical for real-world text data. In this paper we address these challenges by proposing a novel Linearly Interpretable Concept Embedding Model (LICEM) going beyond the current accuracy-interpretability trade-off. LICEM classification accuracy is better than existing interpretable models and matches black-box models. The provided explanations are more plausible and useful with respect to existing solutions, as attested in a user study. Finally, we show our model can be trained without requiring any concept supervision, as concepts can be automatically predicted by the same LLM backbone.

## 1 Introduction

In recent years, Large-Language Models (LLMs) have revolutionized the way we approach text interpretation, generation, and classification (Devlin et al., 2018; Radford et al., 2018; Brown et al., 2020; Achiam et al., 2023; Touvron et al., 2023). Despite their success, LLMs' reliability is insufficient, due to the occurrence of hallucinations (Bang et al., 2023; Huang et al., 2023) and the inconsistency of self-provided explanations that often do not reflect the actual decision-making process (Ye & Durrett, 2022; Madsen et al., 2024; Turpin et al., 2024). Furthermore, traditional explainability methods mainly rely on the attention mechanism (Jain & Wallace, 2019; Wiegreffe & Pinter, 2019) and gradient-based analysis (Chefer et al., 2021b), both of which have been shown to provide limited interpretability as they are often unreliable (Adebayo et al., 2018; Taimeskhanov et al., 2024) and only show *where* the model looks, but not *what* it sees in a given input (Rudin, 2019; Fel et al., 2023; Poeta et al., 2023).

For this reason, Concept-Bottleneck Models (CBMs) (Koh et al., 2020) have been recently proposed in the textual field to improve the interpretability of LLM predictions (Tan et al., 2024b;a). In CBMs, an intermediate layer outputs a set of human-understandable symbols, commonly referred to as concepts, before providing the final classification. Furthermore, CBMs allows concept interventions, i.e., counterfactual predictions based on slight modifications of the predicted concepts. However, they still present several limitations: i) the concept bottleneck architecture prevents to achieve high classification accuracy, particularly in real-world text scenarios where complete concept representations are difficult to obtain; ii) when employing non-linear task predictors, standard CBMs are not *task-interpretable*, i.e., the decision process from the concepts to the final classification is non-interpretable; iii) CBMs concept annotation is expensive and existing generative concept annotation approaches require the employment of multiple modules.

This paper addresses these challenges, by proposing a novel Linearly-Interpretable Concept Embedding Model (LICEM) providing the final prediction in terms of an interpretable linear equation working over concept embeddings. In LICEM both the weights and the concept scores of the linear

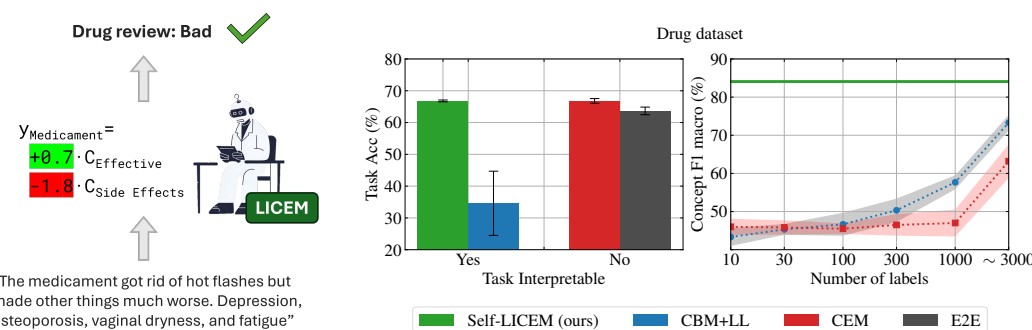

Figure 1: Left, LICEM predicting the sentiment of a drug review (Gräßer et al., 2018). LICEM provides accurate predictions and reveals its decision-making process. Middle, LICEM provides the best accuracy/interpretability trade-off. Right, models' concept F1 scores, when increasing the number of concept annotations. Self-LICEM achieves high scores without requiring concept labels.

equation are predicted for each sample. As shown in Figure 1, left, in the context of a drug review classification, LICEM not only allows identifying the important concepts in the text, such as 'Effective' or 'Side Effects', but also to interpret by-design its local decision process. In the experiment, we positively answer all our research questions. In particular, we show that i) LICEM achieves higher accuracy than existing task-interpretable models while matching or surpassing black-box methods (Figure 1, middle); ii) LICEM explanations are more plausible and useful with respect to existing solutions by means of a user study; iii) LICEM can be trained without any concept annotation (Self-LICEM), as concepts can be automatically predicted by its LLM backbone, providing higher concept accuracy than an existing method (Figure 1, right).

## 2 BACKGROUND

**CBMs.** CBMs (Koh et al., 2020; Tan et al., 2024a) are interpretable models that break the standard end-to-end learning paradigm into the training of two neural modules $f \circ g$. The concept encoder $g : X \to C$ maps raw features $x \in X \subset \mathbb{R}^d$ into $m$ higher-level abstractions $c \in C \subset [0,1]^m$ (i.e., the concepts); the task encoder $f : C \to Y$ predicts $n$ downstream classes based on the learned concepts $\hat{y} = f(g(x)), y \in Y \subset [0,1]^n$. CEMs (Espinosa Zarlenga et al., 2022; Kim et al., 2023) decompose the concept encoder into two functions $g = q \circ h$. The inner function $h : X \to H \subset \mathbb{R}^b$ provides a representation of an input sample, while $q : H \to \mathbf{C}$ maps this representation into $m$ $k$-dimensional concept embeddings $\mathbf{c} \in \mathbf{C} \subset \mathbb{R}^{m,k}$. The concept prediction $\hat{c}_j$ is then given by a neural function over the concept embeddings $\hat{c}_j = s(\mathbf{c_j})$, where $s$ is shared among the $m$ concepts. A probabilistic formalization can be found in Appendix A.1. However, the interpretability of the CEM task predictor $f(\mathbf{c})$ is limited, as the individual dimensions of concept embeddings lack meaningful interpretation. Additionally, adapting CEM to text scenarios remains an open question.

**LLM-based Textual Encoders.** When considering transformer models, there exist several methods for implementing a text encoder $h(x)$. An immediate choice is to employ an encoder-only architecture, such as BERT (Devlin et al., 2018), and extracting the embedding associated to the [CLS] token. However, as recently shown in Jiang et al. (2023b), one can also exploit the remarkable performance of existing decoder-only LLMs. The architecture of an LLM can be conceptualized as comprising two distinct components: the stacked decoder blocks which are responsible for generating a contextualized representation $e$, and a classification head that processes this representation to predict the next token. The first can be interpreted as sampling a representation by the distribution $p_h$, where $h$ is the pre-trained LLM (without the classification head). This distribution can be conditioned toward the generation of specific embeddings by using a prompt $t$, i.e., $e \sim p_h(e|t, x)$. To induce the LLM to generate an embedding which is representative of a sentence, Jiang et al. (2023b) proposed to use the prompt *"this sentence: '[sentence]' means in one word: "* and substituting *'[sentence]'* with the sequence of tokens $x$. In order to obtain a rich representation of the $x$ sequence, exploiting the available knowledge in the pretrained LLM, we thus use the embedding $e \sim p_h(e|t, x)$.

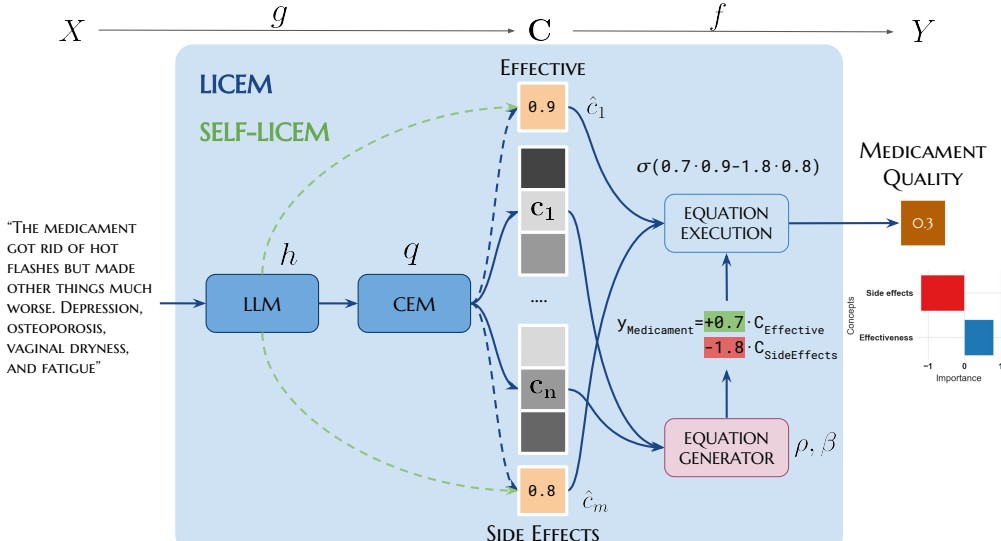

Figure 2: LICEMs visualization. Using a pretrained LLM model, we i) require it to provide an encoding of the input text following; ii) prompt the LLM to generate the concepts predictions $\hat{c}$ (e.g., Side Effects = 0.8) in Self-LICEM, while in LICEM they are provided by a concept embedding layer; iii) make the final prediction in an interpretable way by first predicting the equation weights $w_{ij}$ (e.g., $w_{\text{Side Effects}}$ = -1.8) for predicting the $i$-th class, then executing the resulting linear equation.

## 3 METHOD

In this paper, we aim to develop an interpretable concept-based model for text classification. To achieve this, we need to rely on rich text and concept representations. We create an LLM-based CEM by first using an LLM to model the text encoder and extract an embedding $e$ (as proposed in Jiang et al. (2023b) and discussed in the previous section), which is then fed into a concept embedding layer (Espinosa Zarlenga et al., 2022). Using an LLM as the text encoder allows us to create a powerful task predictor (LICEM, Section 3.1) without the need for fine-tuning the text encoder. Additionally, leveraging pretrained LLMs enables the self-generation of concept predictions (Self-LICEM, Section 3.2), extending the scalability of CBMs to scenarios without available concept annotations. A description of the overall pipeline is provided in Figure 2.

### 3.1 LINEARLY-INTERPRETABLE CONCEPT EMBEDDING MODEL (LICEM)

To create an interpretable predictor, it is essential to utilize both an interpretable data representation and an interpretable model (Ribeiro et al., 2016). Concept-based models allow employing an interpretable data representation within the network. However, to prevent a loss in generalization, CEMs provide the task predictions over concept embeddings whose single dimensions are non-interpretable. Thus, even when using an interpretable task predictor (e.g., a linear layer), CEM does not allow providing an interpretable prediction.

To address this issue, in this work, we propose to *predict a linear equation* that can be executed over the concept predictions and that outputs the final classification as an interpretable aggregation of the most important concepts. In this approach, the neural network's output is modeled as a linear equation where the independent variables are the concepts, and their weights reflect their importance in the task prediction, followed by a bias term. We employ two neural modules to predict *for each sample* the weights and the bias of a linear equation that is executed over the concepts (that are also predicted). Formally:

$$\text{LICEM}: \quad \hat{y}_i = \sigma\left(\sum_j \hat{w}_{ij}\hat{c}_j + \hat{b}_i\right) \qquad \hat{w}_{ij} = \rho_i(\mathbf{c_j}), \ \hat{b}_i = \beta_i(\mathbf{c}), \ \hat{c}_j = s(\mathbf{c_j}) \qquad (1)$$

where, as in common logistic regressions, $\hat{w}_{ij}$ is the weight for the $j$-th concept in predicting the $i$-th task, $b_i$ is the bias for the $i$-th task, while $\hat{c}_j$ and $\mathbf{c_j}$ are, respectively, the prediction and the embedding of the $j$-th concept provided by CEM, and $\sigma$ represents the activation function. For a single concept $j$, the weights for all classes $\hat{w}_j$ are predicted by a neural module $\rho : \mathbf{C_j} \to \mathbb{R}^n$ working on the corresponding concept embedding $\mathbf{c_j}$. As commonly, $\hat{w}_{ij} < 0$ indicates a negatively important concept, $\hat{w}_{ij} > 0$ a positively important one, and $\hat{w}_{ij} \sim 0$ a non-important concept. To improve readability, we aim for sparse weights, where few concepts have $\hat{w}_j \neq 0$. We achieve this by adding $L_1$ regularization to the training loss. The bias term $\hat{b}$ is predicted over all concept embeddings by a function $\beta : \mathbf{C} \to \mathbb{R}$, representing the overall bias for each class. This term is optional, but it allows for positive predictions even when no concept is positively predicted. Indeed, when $\hat{c}_j = 0$ for all $j \in \{1, ..., m\}$, the prediction would be $\hat{y}_i = 0$ regardless of $\hat{w}_{ij}$. To prevent over-reliance on the bias term, we add $L_2$ regularization to encourage small bias values, minimizing its influence on task prediction. Finally, we use a sigmoid activation function $\sigma$ for binary classification tasks and a softmax for multi-class classification tasks. To understand the contribution of a concept to the final prediction of a class, we propose considering the combined contribution $\hat{w}_{ij}\hat{c}_j$ and plotting them in a LIME-like feature importance plot, as shown in the output of Figure 2.

**Training.** LICEM is trained similarly to any supervised concept-based model with a cross-entropy $H$ loss over both the predicted concepts and the tasks:

$$\mathcal{L}_{sup} = H(c, \hat{c}) + \lambda_y H(y, \hat{y}) + \lambda_w ||w||_1 + \lambda_b ||b||_2 \qquad (2)$$

where we indicate the loss over the concept predictions as $\mathcal{L}_c = H(\hat{c}, c)$, the loss over the task predictions as $\mathcal{L}_t = H(\hat{y}, y)$, with $||w||$ and $||b||$ the regularization terms over the weights and biases and with $\lambda_y, \lambda_w$ and $\lambda_b$ the optimization weights for each term. In the rest of the paper, we will refer to this strategy as *supervised*.

## 3.2 Exploiting LLMs to avoid concept annotation: a Self-Generative approach

To alleviate human annotators from the burden of providing concept supervision, a few works are starting to exploit the knowledge already available in pre-trained LLMs, both in the image (Yang et al., 2023; Oikarinen et al., 2023) and in the textual domains (Ludan et al., 2023). First, an LLM is asked to provide several attributes that describe each class. Each attribute is considered a concept for that class, possibly shared with other classes. E.g, a *parrot* may be described as being a bird, with *bright feathers* and of *medium size*. Then another LLM is required to predict whether the concept is present in the input samples. The LLM, in this case, is formally represented by the distribution $p_\theta$, where $\theta$ denotes the parameters of a pre-trained LLM with classification head. When conditioned on a prompt $t$, the model generates the token "yes" if a specific concept is identified in the input text sequence $x$, and "no" otherwise. Thus, the predicted concept is sampled as $c' \sim p_\theta(c'|t, x)$. In Appendix A.2 we report some examples of prompts.

**Generative approach.** In Ludan et al. (2023), these concept predictions $c'$ are used as labels to train a textual concept encoder. Formally, $\mathcal{L}_{gen} = \mathcal{L}_{c'} + \lambda \mathcal{L}_t = H(c', \hat{c}) + \lambda H(y, \hat{y})$. We will refer to this strategy as *generative*, as a generative model provides concept annotations.

**Self-generative approach.** While the generative approach reduces human annotation efforts, it requires training an additional concept encoder to learn the LLM-provided labels. In this paper, since we already employ an LLM as a text encoder, we propose using the same LLM to directly make the concept predictions. More precisely, we prompt the LLM to provide both a representation $e$ for each sample $x$ and the concept predictions, i.e., $\hat{c} = c' \sim p_\theta(c'|t, x)$. This results in a modification of both CEM and LICEM as the concept predictions are self-generated by the same LLM, as shown in Figure 2. We will refer to this approach as *self-generative*, as the same model directly provides the concept predictions. This method eliminates the need for concept annotations, but also reduces the number of parameters to train and improves concept performance if compared to the generative method. Indeed, the concept accuracy of the self-generative method represents an optimum for the generative one. In the former, the concepts $c'$ provided by the LLM are directly used as concept predictions, while in the latter, they serve as training labels for an external text encoder, which aims to replicate $c'$. Self-LICEM is obtained by substituting the concept predictions $\hat{c}$ with $c'$ from Equation 1:

$$\text{Self-LICEM} \quad \hat{y}_i = \sigma \left( \sum_j \hat{w}_{ij} c'_j + \hat{b}_i \right). \qquad (3)$$

The concept embedding encoder $q$ and the neural modules $\rho$ and $\beta$ producing the interpretable linear equation are trained as in Equation 2, but minimizing, this time, only the loss over the task:

$$\mathcal{L}_{selfgen} = H(y, \hat{y}) + \lambda_w ||w||_1 + \lambda_b ||b||_2, \tag{4}$$

This approach is not limited to LICEM; it can also be extended to CBM-based and CEM-based models. In these cases, the LLM provides the concept predictions (CBM) or both the predictions and the embedding (CEM). In both cases, the optimization strategy involves minimizing only the cross-entropy on the task predictions $H(y, \hat{y})$, as shown in Eq. 4. This allows converting any pre-trained LLM into a concept-based model without the need for concept annotations.

## 4    EXPERIMENTS

In this section, we want to answer the following research questions:

- **Generalization.** Does LICEM achieve superior performance in text classification compared to other interpretable models, and is it on par with non-interpretable ones? (Section 4.2)

- **Concept Efficiency.** How many concept supervisions are required to match Self-LICEM accuracy? Does the self-generative strategy outperform the generative one in concept accuracy? (Section 4.3)

- **Interpretability.** Are LICEM explanations more interpretable than those of other methods? Can we effectively interact with LICEM? (Section 4.4)

### 4.1    SETUP

We test LICEM performance over different datasets (both with and without concept-supervisions), comparing against several models and for different metrics. For all experiments, we report the average and standard deviation across three repetitions. The models were trained on a dedicated server equipped with an AMD EPYC 7543 32-Core processor and one NVIDIA A100 GPU. Our code is publicly available at `www.example.com`[1]

**Dataset.** We evaluated LICEM performance on three text-classification datasets for which concept annotation is available: CEBaB (Abraham et al., 2022), MultiEmotions-IT (Sprugnoli et al., 2020), and Drug review (Gräßer et al., 2018). Additionally, we tested the generative and self-generative approaches on the Depression dataset (Yates et al., 2017), where concept annotations are unavailable, but where an LLM (Jiang et al., 2024) identified six depression-related concepts which are: 'Self-deprecation', 'Loss of Interest', 'Hopelessness', 'Sleep Disturbances', 'Appetite Changes', and 'Fatigue'. Further information regarding the datasets is reported in Appendix A.3.

**Baselines.** We compare LICEM against several baselines, including black-box and concept-based models, both task-interpretable and non-interpretable approaches. For all models, we use a non fine-tuned Mixtral 8x7B (Jiang et al., 2024) encoder $h(x)$, following the encoding strategy proposed in Jiang et al. (2023b). In Appendix A.4 we also report all results based on a fine-tuned BERT encoder (Devlin et al., 2018) as backbone. The results show that the decoder-only LLM achieves similar performance without fine-tuning the whole LLM. Besides, it enables the self-generative approach: in Appendix A.5 we report a comparison of the concept annotation performance when using different LLMs. For black-box models (E2E), we evaluate an end-to-end model directly classifying the task with a Mixtral encoder $h(x)$ and few layers as classification head (MLP), and the same Mixtral used in Zero-shot and Few-shot prompting. CBM+LL and CBM+MLP are the two CBMs originally proposed in (Koh et al., 2020) and recently adapted to text in (Tan et al., 2024b). They employ a concept bottleneck layer followed, the first one, by an interpretable linear layer, while the second by a non-interpretable multi-layer perceptron. CBM+DT and CBM+XG are respectively two CBM variants proposed in (Barbiero et al., 2023), using an interpretable decision tree and a non-interpretable XGBoost classifier (Chen & Guestrin, 2016) on top of the concept bottleneck layer, respectively. CBM+DT is task-interpretable, as one can extract a decision rule based on concepts, whereas the second variant CBM+XG is non-interpretable. As described in Section 2, CEM (Espinosa Zarlenga et al., 2022) employs embeddings to represent concepts and enhance CBM

---

[1]We will release the code upon paper acceptance.

Table 1: Task accuracy (%) of the compared models. We report in **bold** the best result among the same type of models (e.g., supervised, interpretable ones) considering models equally best if their standard deviations overlap. We use ✓ to indicate models requiring concept supervision (C. Sup.) or having a task-interpretable predictor (T. Inter.). We highlight in light gray the models we propose in this work. The Self-Generative approach extends the scalability of concept-based models to datasets without concept annotations, where supervised models cannot be applied ($-$).

| Type | Method | C. Sup. | T. Inter. | CEBaB | Multiemo-It | Drug | Depression |
|------|--------|---------|-----------|-------|-------------|------|------------|
| E2E | Mixtral–MLP | ✗ | ✗ | **88.80** $\pm$ 0.75 | 80.01 $\pm$ 0.63 | **63.66** $\pm$ 1.20 | **97.18** $\pm$ 0.03 |
| | Mixtral–Zero-shot | ✗ | ✗ | 86.80 $\pm$ 0.31 | 80.06 $\pm$ 0.66 | 60.81 $\pm$ 0.28 | 73.77 $\pm$ 0.23 |
| | Mixtral–Few-shot | ✗ | ✗ | 84.79 $\pm$ 0.42 | **84.17** $\pm$ 0.67 | 62.16 $\pm$ 0.27 | 76.38 $\pm$ 0.08 |
| SUP. | CBM+MLP | ✓ | ✗ | 78.41 $\pm$ 9.30 | 45.43 $\pm$ 8.20 | 45.42 $\pm$ 4.90 | $-$ |
| | CBM+XG | ✓ | ✗ | 83.01 $\pm$ 0.10 | 69.01 $\pm$ 0.02 | 55.00 $\pm$ 0.13 | $-$ |
| | CEM | ✓ | ✗ | **89.60** $\pm$ 0.49 | **83.33** $\pm$ 0.47 | **66.81** $\pm$ 0.40 | $-$ |
| | CBM+LL | ✓ | ✓ | 71.43 $\pm$ 9.71 | 42.67 $\pm$ 7.01 | 34.60 $\pm$ 10.10 | $-$ |
| | CBM+DT | ✓ | ✓ | 77.20 $\pm$ 0.40 | 65.00 $\pm$ 0.02 | 47.20 $\pm$ 0.40 | $-$ |
| | DCR | ✓ | ✓ | 88.05 $\pm$ 0.53 | 82.01 $\pm$ 0.71 | 65.40 $\pm$ 0.80 | $-$ |
| | LICEM (ours) | ✓ | ✓ | **89.89** $\pm$ 0.77 | **83.47** $\pm$ 0.49 | **66.80** $\pm$ 0.29 | $-$ |
| SELF GEN. (OURS) | Self-CBM+MLP | ✗ | ✗ | 82.71 $\pm$ 0.01 | 75.42 $\pm$ 4.42 | 47.59 $\pm$ 0.33 | 82.31 $\pm$ 0.04 |
| | Self-CBM+XG | ✗ | ✗ | 82.70 $\pm$ <0.01 | 79.09 $\pm$ <0.01 | 53.28 $\pm$ <0.01 | 82.28 $\pm$ <0.01 |
| | Self-CEM | ✗ | ✗ | **89.14** $\pm$ 0.38 | **84.06** $\pm$ 0.09 | 65.20 $\pm$ 0.73 | **97.16** $\pm$ 0.08 |
| | Self-CBM+LL | ✗ | ✓ | 82.71 $\pm$ 1.23 | 77.15 $\pm$ 0.96 | 47.35 $\pm$ 0.29 | 82.12 $\pm$ 0.15 |
| | Self-CBM+DT | ✗ | ✓ | 83.95 $\pm$ <0.01 | 78.44 $\pm$ <0.01 | 53.28 $\pm$ <0.01 | 82.28 $\pm$ <0.01 |
| | Self-DCR | ✗ | ✓ | 87.72 $\pm$ 0.66 | 83.47 $\pm$ 0.43 | 63.29 $\pm$ 0.36 | 97.11 $\pm$ 0.03 |
| | Self-LICEM | ✗ | ✓ | **89.56** $\pm$ 0.29 | **84.49** $\pm$ 0.25 | **65.89** $\pm$ 0.39 | **97.23** $\pm$ 0.02 |

generalization performance, but at the cost of losing task interpretability. Finally, DCR (Barbiero et al., 2023) is a neuro-symbolic approach designed to improve the interpretability of CEM. It generates propositional rules executed by a fuzzy system on top of concept predictions. We adapt CEM and DCR to work in the text classification scenario, and we compare their performance against the proposed model. For the training details regarding each model, please refer to Appendix A.3.

**Metrics.** We evaluate LICEM using various metrics. To assess **generalization** performance, we compute the task accuracy and the macro-averaged concept F1 score (as concept classes are highly imbalanced); for self-generative models, the macro-averaged F1 score evaluates the concept predictions directly provided by the LLM (Section 3.2). To measure **efficiency**, we examine the concept F1 score of all models when increasing the number of concept annotations. For **interpretability**, we first evaluate LICEM explanations through a user study, comparing their plausibility and usefulness to that of DCR; secondly we evaluate the effectiveness of concept interventions over LICEM to enhance classification accuracy (Espinosa Zarlenga et al., 2024); third we measure the Causal-Concept Effect (CaCE) (Goyal et al., 2019), which assesses the causal relevance of concepts for task predictions.

## 4.2 LICEM GENERALIZATION (TABLE 1)

**LICEM matches black-box task performance and outperforms all task-interpretable models.** The initial finding from analyzing Table 1 is that LICEM consistently delivers task performances that are comparable or even better than black-box and non task-interpretable models. Interestingly, although with overlapping standard deviation, E2Es are never the best performing models, which is three times a LICEM and once a CEM. When it comes to interpretable models, LICEM invariably emerge as the best interpretable predictor among the compared, with an improvement of at least 7-19% over CBM interpretable variants. With respect to DCR, we believe the improvement is due to the way LICEM provide the final classification: the parameters of a linear equation are easier to predict than constructing a logic rule, and to optimize, as they do not require passing through a fuzzy system.

**Self-generative approach increasing CBMs scalability while maintaining task performance.** Self-generative CBMs maintain the task accuracy of supervised CBMs while increasing their scalability, as they can be applied to scenarios where concept annotations are not available, such as

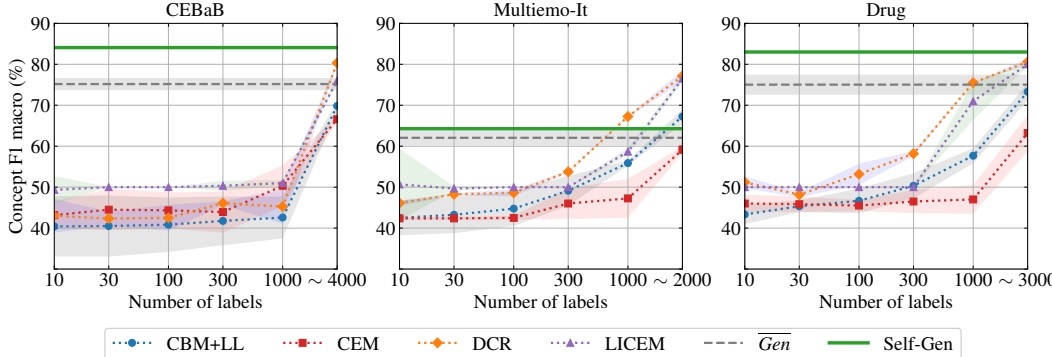

Figure 3: Concepts prediction performance vs number of concept labels used during training. To increase plot readability, we only included the CBM+LL and the average F1 score for the generative approaches ($\overline{Gen}$). Self-Gen. and Gen. approaches are reported with a straight line, as they do not require concept annotation.

the Depression dataset. We reported the performance of all concept-based baselines (not only Self-LICEM) when trained along the self generative approach to show that it enables all CBMs to work on top of a pretrained LLM without concept annotations. When comparing the model performance in the two approaches, we can generally notice that the confidence intervals are overlapping. In a few cases, such as CBM+LL, we can notice a stable improvement over all the datasets when using the self-generative approach up to $+30\%$ on the Multiemo-It dataset. This is likely due to the trade-off posed when training a concept-bottleneck layer, which has to favor either the task or the concept performance: when directly working over good concept predictions, CBM performance improves. In Appendix A.6, we also report the task accuracy of models trained along the standard generative approach, showing similar results.

## 4.3 LICEMs concept efficiency (Figure 3)

**Self-Generative approach strongly reduces the human annotation effort.** In Figure 3, we report the concept prediction performance of the compared methods when increasing the number of concept labels used for training. Self-generative and generative approaches are reported with a straight line since they do not require any concept supervision[2]. Generative and self-generative models achieve a concept macro-averaged F1 score that is higher or close to that of supervised models when using all available annotations, and significantly higher otherwise. When considering the CEBaB and Drug datasets, supervised models do not surpass Self-Gen even when using all concept annotations, with the latter achieving the highest concept accuracy. Likely, the amount of concept annotations required to match the accuracy of the self-generative approach exceeds what is available in these datasets.

**The self-generative concept accuracy exceeds that of the generative approach.** The concepts prediction performance of the generative approach tends to be lower than that of the self-generative approach, with a reduction ranging from 2% to 7% in F1 macro score. This is because the concepts predicted by generative models are approximations of the self-generated concepts $c'$ used in the self-generative approach. These self-generated concepts serve as the labels for training the concept encoders in the generative learning process. Detailed concepts prediction performance is presented in Appendix A.6, Table 6 for all models across all datasets, when provided with full concept annotations.

## 4.4 LICEM interpretability

**LICEM explanations are more plausible and more useful than DCR (Fig. 4).** To evaluate the interpretability of LICEM explanations, we conducted a user study comprising 21 ques-

---

[2]Generative approaches results are reported with variance because the concepts are still learnt and thus the performance vary across models. For the self-generative approach, instead, the result does not vary because the concepts are predicted equally by the LLM for all models since we set the LLM's temperature to zero, which results in a deterministic annotation.

tions and involving 46 participants, consisting of both machine learning experts and non-experts. It is structured as follows. First, participants are asked to choose the most plausible explanation (Rajagopal et al., 2021) from three options: the LICEM explanation, the DCR explanation, or neither. This process is repeated across three datasets (we excluded Multiemo-It, as it contains only Italian comments). Examples of the questions are shown in Figure 6, 7, and 8. In a second task, we assess explanation usefulness by computing how much participants can guess the model predictions based on the provided explanation (Fel et al., 2023).

This experiment is carried out for both LICEM and DCR explanations and is repeated across the same datasets as in the previous step. In both cases, the samples have been randomly drawn from each dataset. A complete characterization of the user study is reported in Appendix A.7. The left image of Fig. 4 presents the results related to explanation plausibility. It is evident that the LICEM explanation is consistently considered more plausible over the rule-based DCR explanation by both expert and non-expert users. Contrary to our expectations, LICEM was espe-

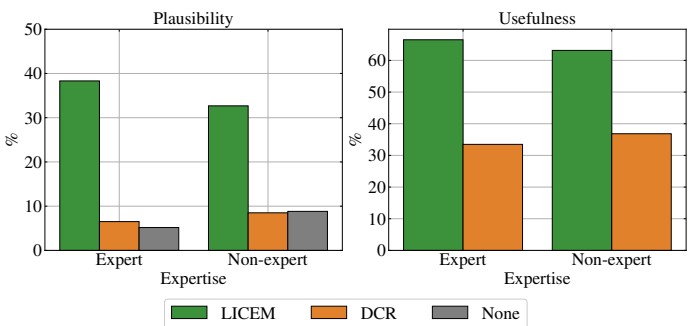

Figure 4: Averaged survey results for the two user groups. On the left, we report the explanation plausibility; on the right, users' accuracy in guessing the model prediction based on its explanation.

cially favored by expert users, with nearly 80% of them appreciating its explanations. The right image of Fig. 4 illustrates the accuracy achieved by users when tasked with selecting a class label based on a given explanation. Both groups of users demonstrated good accuracy when making classifications using the LICEM explanations. Expert people, in particular, nearly double the accuracy when using LICEM compared to when using DCR explanations.

**LICEM is responsive to concept interventions (Figure 5).** To assess the possibility to interact with LICEM, we evaluated the effect of concept interventions, i.e., modifications at test time of the predicted concepts with a concept provided by a human expert. Figure 5 shows the test task accuracy gain with increasing intervention probability on the CEBaB dataset, demonstrating LICEM's responsiveness and significant performance improvement. A similar behaviour can also be observed for CBMs, even though they were starting from a lower task accuracy and a higher increase could have also been expected. Results for all datasets are reported in Appendix A.8, showing similar results, with LICEM always improving its task accuracy through interactions. For comparison, we also report the E2E model with a flat line, since it does not offer this possibility.

**LICEM predictions are caused by most important concepts (Table 2).** We assess the responsiveness of concept-based models to *do-interventions* over concepts (Pearl et al., 2016), by computing the causal concept effect (CaCE) (Goyal et al., 2019). CaCE measures the impact of modifying input

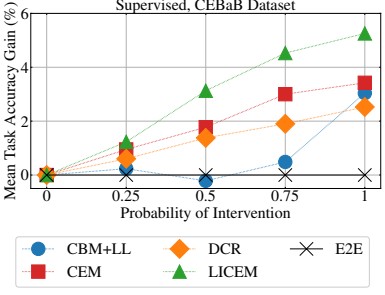

Figure 5: Concept interventions on the CE-BaB dataset. We report the task accuracy gain when varying the probability of intervention.

Table 2: Causal Concept Effect (CaCE) for different methods. A high (absolute) value implies a strong responsiveness of a model to modifications to the concept.

| Model | Food | Amb. | Service | Noise |
|-------|------|------|---------|-------|
| CBM+LL | -0.02 | 0.01 | 0.01 | -0.01 |
| CEM | 0.29 | 0.08 | 0.13 | -0.05 |
| DCR | 0.33 | 0.02 | 0.20 | -0.02 |
| LICEM | 0.62 | 0.18 | 0.37 | 0.15 |
| Self-LICEM | 0.63 | 0.20 | 0.35 | 0.15 |

samples on model predictions. For concept-based models, interventions can be made at the concept level (Dominici et al., 2024). In the evaluated dataset, several concepts are globally relevant for task classification (positively or negatively), thus we expect models to exhibit high absolute CaCE values. In Table 2, we report the results for the CEBaB dataset: both LICEMs demonstrate high CaCE values, particularly for 'Food' and 'Service' which are crucial concepts. These values are higher than CEM and DCR, suggesting a stronger reliance on the prediction over these concepts. Conversely, CBMs report low CaCE values that may indicate concept leakage issues (Marconato et al., 2022), possibly due to the constraints of the concept-bottleneck representation. Results for all datasets are reported in Appendix A.9, showing consistent findings.

## 5 RELATED WORK

**LLM interpretability.** Recent studies have highlighted the unreliability of LLMs, as they often occur hallucinations (Ji et al., 2023), and when prompted for explanations, their responses frequently do not reflect the actual decision-making process (Ye & Durrett, 2022; Madsen et al., 2024; Turpin et al., 2024). Although the attention mechanism in transformer models offers some interpretability, it has been criticized for its lack of clarity and consistency (Jain & Wallace, 2019; Wiegreffe & Pinter, 2019). To improve LLM explainability, various standard XAI techniques, such as LIME (Ribeiro et al., 2016) and Shapley values (Lundberg & Lee, 2017), along with newer methods (Kokalj et al., 2021; Heyen et al., 2024; Chefer et al., 2021b;a), have been employed. However, these standard techniques have limitations (Kindermans et al., 2019; Ghorbani et al., 2019; Adebayo et al., 2018; Taimeskhanov et al., 2024), primarily because they explain predictions in terms of input features that often lack meaningful interpretations for non-experts (Poursabzi-Sangdeh et al., 2021). Consequently, researchers are now exploring interpretable-by-design models also in the textual domain (Rajagopal et al., 2021; Jain et al., 2022; Tan et al., 2024b;a).

**Concept-based models.** Concept-based models (Alvarez Melis & Jaakkola, 2018; Koh et al., 2020; Ciravegna et al., 2023; Kim et al., 2023) are transparent and interactive models that utilize an intermediate layer to represent concepts. To increase the representation capability of the concept layer, Espinosa Zarlenga et al. (2022) proposed using concept embeddings. However, the interpretability of CEM task predictor is limited, as individual embedding dimensions lack clear meaning. In this work, we demonstrate how to create an interpretable task predictor over these embeddings. A recent neurosymbolic method (DCR, Barbiero et al. (2023)) based on fuzzy logic attempted to tackle this issue. We extend CEM and DCR applicability to the textual domain, while showing that LICEM achieves superior predictive performance than DCR and higher interpretability than both. Additionally, supervised concept-based models (Koh et al., 2020; Espinosa Zarlenga et al., 2022) often require extensive concept annotations, which are frequently unavailable, particularly in text. We enhance a recent generative approach (Yang et al., 2023; Oikarinen et al., 2023; Ludan et al., 2023) by using the same LLM for self-generated concept predictions and sample representations.

## 6 CONCLUSION

In this paper, we propose LICEM, a novel linearly interpretable concept-based model for text classification. The experimental results show this model matches black-box models performance, is interpretable and can be trained without concept supervision (Self-LICEM). Besides a technological impact, we believe this work can also positively impact the society by enhancing LLM transparency and interpretability, thus facilitating their employment in several fields such as Healthcare, Finance, Legal Systems and Autonomous Vehicles.

**Future work.** In this analysis, we focus on binary or ternary sentiment analysis for the ease of identifying concepts, and to texts composed of a few sentences. In future work, we will extend our analysis to other NLP tasks and to longer texts, to ensure the scalability of this approach. Specifically, we plan to extend the capability of this model to work in language modelling tasks, similarly to Ismail et al. (2023) employing CBMs to solve generative tasks in computer vision. Furthermore, other interpretable functions could be generated and used to provide an interpretable prediction, besides linear equations. As an example, we could also generate a text describing how each concept has been predicted and its role in the final prediction, together with the indication of the task prediction. We leave these investigations for future research.

**Text**: what a perfect spot for a romantic dinner; amazing service; and wonderful food. super quiet too!
**Label**: Positive

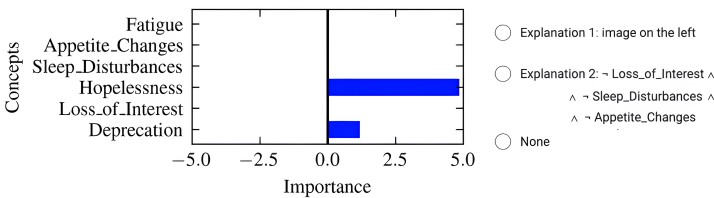

Figure 6: Example of explanation plausibility question, CEBaB dataset.

**Text**: sometimes i think i wa either born too early or too late for my life the shape of water anyone else feel this way sometimes i feel like somehow i wasn t supposed to be here i don t seem to fit in with my life finding someone i click with ha become like finding life on other planet at it s difficult not to momentarily succumb to feeling of quiet heart heavy despair.
**Label**: Depressed

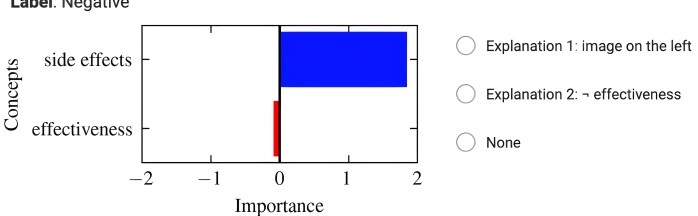

Figure 7: Example of explanation plausibility question, Depression dataset.

**Text**: infection cleared up but the cough did not. the bronchitis returned and is still present. the z pack was used on 2 seperate occasions but after the cough mucus was back to a clear color, my physician said i was fine [...] the side effects were stomach cramps and diarrhea for 2 weeks [...]
**Label**: Negative

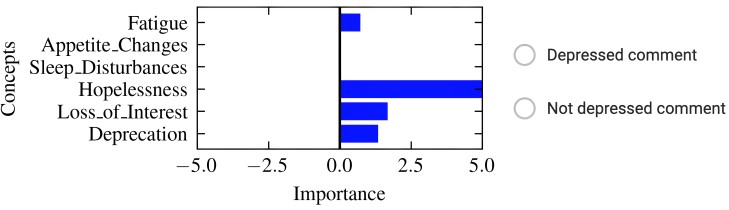

Figure 8: Example of explanation plausibility question, Drug dataset.

According to the explanation, select the correct label.

Figure 9: Example of explanation-based prediction, Depression dataset.

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

# A APPENDIX / SUPPLEMENTAL MATERIAL

## A.1 PROBABILISTIC FORMALIZATION

**CBMs.** As stated in Section 2, CBMs (Kim et al., 2023) provide explanation operating abstract human-understandable concepts. Let $x \sim p(x)$ represent the random variable drawn from the data distribution, and $c_j \sim p(c_j|x)$ denote the concept $j$ derived from sample $x$, where $c_j \in [0, 1]$. The prediction of the categorical variable $y$ is then formulated as:

$$y \sim p(y|x) = p(y|c_1, ...c_m) \underbrace{\prod_{j=1}^{m} p(c_j|x)}_{p(c_1, ..., c_m|x)} \tag{5}$$

where $p(y|c_1, ...c_m)$ is a categorical distribution usually modeled using a fully connected layer, and $\prod_{j=1}^{m} p(c_j|x)$ is parameterized by a neural model. For simplicity we define $p(y|c_1, ..., c_m) = p(y|c)$ and $p(c|x) = p(c_1, ..., c_m|x)$. The different components are optimized by maximizing the following loss function:

$$\mathcal{L} = E_{x,c \sim p(x,c)}[-\log p(c|x)] + \lambda_y E_{x,y \sim p(x,y)}[-\log p(y|x)] \tag{6}$$

where $\lambda_y \in [0, 1]$ is the coefficient used to prioritize the concept learning relative the task learning.

**CEMs.** CEMs (Espinosa Zarlenga et al., 2022) addresses the low task performance of CBMs, which is attributed to the bottleneck created by the intermediate concept layer, generating a concept embedding for each concept. Initially, a vector representation of the raw data is generated, denoted as $e \sim p(e|x)$, where $e \in R^l$. Subsequently, both the active and inactive concept states, represented as $\mathbf{c}_j^+, \mathbf{c}_j^- \in R^k$, are derived from the two conditional distributions $p(\mathbf{c}_j^+|e)$ and $p(\mathbf{c}_j^-|e)$. At this stage the concept score $\hat{c}_j \in [0, 1]$ is sampled as $\hat{c}_j \sim p(\hat{c}_j|\mathbf{c}_j^+, \mathbf{c}_j^-)$. The representational embedding of concept $j$, denoted as $\mathbf{c}_j \in R^k$, is computed as a convex combination of the active and inactive states, given by $\mathbf{c}_j = \hat{c}_j \cdot \mathbf{c}_j^+ + (1 - \hat{c}_j) \cdot \mathbf{c}_j^-$. Finally, all concept embeddings are utilized to condition the generation of the target variable, expressed as $y \sim p(y|\mathbf{c}_1, ..., \mathbf{c}_m)$. This process can then be formalized by

$$y \sim p(y|x) = \underbrace{p(y|\mathbf{c}_1, ..., \mathbf{c}_m)}_{\text{CLASSIFIER}} \prod_{j=1}^{m} \underbrace{\left[ p(\hat{c}_j|\mathbf{c}_j^+, \mathbf{c}_j^-)p(\mathbf{c}_j^+|e)p(\mathbf{c}_j^-|e) \right]}_{\text{CEM}} \underbrace{p(e|x)}_{\text{ENCODER}} \tag{7}$$

The classifier, which operates on the concatenation of concept embeddings, is typically structured as a deep neural network to enable end-to-end optimization. The loss function to optimize is analogous to 6.

**LICEMs.** LICEM builds over the CEM's output, modeling the distribution associated to the classifier in 7. It utilizes both concept embeddings and concept scores to generate explanations, representing them as a linear combination of concepts, were the weight associated to each concept $j$ regarding class $i$, $\hat{w}_{ij} \sim p_{\rho_i}(\hat{w}_{ij}|\mathbf{c_j})$, changes according to the concept embedding. Additionally, a dynamic bias is sampled using all the concept embeddings $\hat{b}_i \sim p_{\beta_i}(\hat{b}_i|\mathbf{c}_1, ..., \mathbf{c}_m)$. The logit corresponding to class $i$ is calculated as $l_i = \sum_j \hat{w}_{ij}\hat{c}_j + \hat{b}_i$. For a multiclass classification task, the softmax function is applied to the computed logits. The final probability associated with class $i$ is given by $\pi_i = Softmax(l_i)$. The predicted class label $\hat{y}$ is subsequently sampled from a categorical distribution defined by $\hat{y} \sim Cat(\hat{y}|\pi_1, ..., \pi_n)$, where $n$ denotes the total number of classes.

**Self-generative.** With the self-generative approach CEM is modified in order to allow external concept scores injection. This traduces into eliminating the neural module which models $p(\hat{c}_j|\mathbf{c}_j^+, \mathbf{c}_j^-)$, and using the LLM generated scores $\hat{c}_j \sim p(c_j|x, t)$, where $t$ represents the prompt, to select the state of the concept embedding $\mathbf{c}_j = \hat{c}_j \cdot \mathbf{c}_j^+ + (1 - \hat{c}_j) \cdot \mathbf{c}_j^-$. Using $\mathbf{c} = (\mathbf{c}_1, ..., \mathbf{c}_m)$ and $\hat{c} = (\hat{c}_1, ..., \hat{c}_m)$ to simplify the notation, the label prediction process can be formalized as:

$$y \sim p(y|x) = \underbrace{p(y|\mathbf{c}, \hat{c})}_{\text{LICEM}} \prod_{j=1}^{m} \underbrace{\left[ p(\mathbf{c}_j|e, \hat{c}_j) \right]}_{\text{MODIFIED CEM}} \underbrace{p(\hat{c}|t, x)}_{\text{LLM}} \underbrace{p(e|x)}_{\text{ENCODER}} \tag{8}$$

## A.2 PROMPTS FOR ANNOTATION

Here we report the prompts used to instruct *Mistral 7B* and *Mixtral 8x7B* to perform the annotations on the 4 different datasets used in this work. We adopted the in-context instruction learning prompting strategy (Ye et al., 2023).

### CEBAB

```
In a dataset of restaurant reviews there are 4 possible concepts: Good Food, Good Ambiance, Good Service and
Good Noise. Given a certain review, you have to detect if those concepts are present or not in the review.

Answer format: Good Food:score, Good Ambiance:score, Good Service:score, Good Noise:score.

Do not add any text other than that specified by the answer format.
The score should be equal to 1 if the concept is present or zero otherwise, no other values are accepted.

The following are examples:

Review: "The food was delicious and the service fantastic".
Answer: Good Food:1, Good Ambiance:0, Good Service:1, Good Noise:0

Review: "The staff was very rough but the restaurant decorations were great. Other than that there was a very
relaxing background music".
Answer: Good Food:0, Good Ambiance:1, Good Service:0, Good Noise:1

Now it's your turn:

Review: <review>
Answer:
```

### DRUG

```
In a dataset of drug reviews there are 2 possible concepts:

- Effectiveness: 1 if the drug was highly effective and 0 if it was marginally or not effective,
- Side effects: 1 if the drug gave side effects and 0 otherwise.

Given a certain review, you have to detect if those concepts are present or not in the review.

Answer format: Effectveness:score, Side effects:score.

Do not add any text other than that specified by the answer format.
The score should be equal to 1 if the concept is present or zero otherwise, no other values are accepted.

The following are examples:

Review: "The medicine worked wonders for me. However, I did experience some side effects. Despite this,
I still found it easy to use and incredibly effective".
Answer: Effectiveness:1, Side effects:1

Review: "Not only it did fail to alleviate my symptoms, but it also led to unpleasant side effects".
Answer: Effectiveness:0, Side effects:1

Now it's your turn:

Review: <review>
Answer:
```

### MULTIEMO-IT

```
In a dataset containing comments in Italian, you need to identify the following concepts:

-Joy: the user who wrote the comment expresses joy,
-Trust: the user who wrote the comment expresses trust,
-Sadness: the user who wrote the comment expresses sadness,
-Surprise: the user who wrote the comment is surprised.

Response format: Joy:score, Trust:score, Sadness:score, Surprise:score.
```

```
The score must be equal to 1 if the concept is present and 0 otherwise; other values are not accepted.

The following is an example:
Comment: "Mi piace la rivisitazione di questa canzone, dolce, raffinata, elegante, bellissima!"
Answer: Joy:1, Trust:1, Sadness:0, Surprise:1

Now it's your turn:
Comment: <comment>
Answer:
```

DEPRESSION

```
You have to identify the presence or absence of 6 concepts in a given text.
The concepts to be identified are:

- Self-Deprecation: the text exhibits self-critical or self-deprecating language, expressing feelings
of guilt, shame, or inadequacy.
- Loss of Interest: diminished pleasure or motivation in the writer's descriptions of hobbies or pursuits.
- Hopelessness: the writer express feelings of futility or a lack of optimism about their prospects.
- Sleep Disturbances: the writer mentions insomnia, oversleeping, or disrupted sleep as part of their
experience.
- Appetite Changes: there are references to changes in eating habits.
- Fatigue: there are references to exhaustion or lethargy.

Answer format: Self-Deprecation:score, Loss of Interest:score, Hopelessness:score, Sleep Disturbances:score,
Appetite Changes:score, Fatigue:score.

The score has to be 1 if the concept is detected and 0 otherwise. Do not add any other text besides the one
specified in the answer format.

Text: <text>
Answer:
```

## A.3 EXPERIMENTAL DETAILS

**Dataset**   To check the performance of LICEM, we first selected three text-classification datasets for which concept annotations are provided or in which attribute annotations can be employed. The first dataset is CEBaB (Abraham et al., 2022), a dataset designed to study the causal effects of real-world concepts on NLP models. It includes short restaurant reviews annotated with sentiment ratings at both overall-review level and for four dining experience aspects (food quality, noise level, ambiance, and service). The second dataset is MultiEmotions-IT (Sprugnoli et al., 2020), a dataset designed for opinion polarity and emotion analysis and containing comments related to videos and advertisements posted on social media platforms. These comments have been manually annotated according to different aspects, among which we choose two dimensions: opinion polarity, describing the overall sentiment expressed by users (that we employed as task labels), and basic emotions from which we selected joy, trust, sadness, and surprise (concept labels). The third dataset is Drug review (Gräßer et al., 2018), a dataset that provides patient reviews on specific drugs. The reviews are annotated with the overall satisfaction of the users (which we discretize to a binary representation) and drug experience annotations as effectiveness and side effects. Furthermore, to test the generalization capability of self-supervised methods in a scenario where concept annotations are not actually provided, we chose the Depression dataset (Yates et al., 2017)[3] which consists of Reddit posts for users who claimed to have been diagnosed with depression and control users. The set of concepts utilized for the Depression dataset was generated by the same LLM employed for the annotations, *Mixtral 8x7B* (Jiang et al., 2024). Upon prompting the model to identify concepts relevant to depression-related comments, it returned the following six key concepts: self-deprecation, loss of interest, hopelessness, sleep disturbances, appetite changes, and fatigue.

**Evaluation**   We evaluate LICEM against the baselines according to different metrics, each one analysing a different characteristic of the models. First, to check LICEM **generalization** performance, we compute the task accuracy and the macro-averaged F1 score for concepts prediction. For GENERATIVE and SELF-SUP methods, we train the model without employing the actual concept annotations but by prompting an LLM as described in Section 3.2. To test the **efficiency** of the models, we report the concepts prediction performance of the models when increasing the number of

---

[3]For the Depression dataset, we employed the cleaned version available on Kaggle.

concept annotations (provided by humans). Finally, to test the **interpretability** of the model, we first conducted a user study involving 30 participants, consisting of both machine learning experts and non-experts to evaluate LICEM explanations. Secondly, we checked whether it is possible to intervene on the predicted concepts (Espinosa Zarlenga et al., 2024) and improve the classification accuracy even when using an interpretable predictor. Thirdly, we checked the Causal-Concept Effect (CaCE) (Goyal et al., 2019), a measure introduced to assess the causality of a model with respect to a given concept. Concept-based models, indeed, are generally required to make task predictions according to the predicted concepts. However, the employment of vectorial concept representations (Mahinpei et al., 2021) may lead to model ignoring the predicted concepts. We see in the results that this is not the case for LICEM.

**Experimental settings**    For the E2E, CBMs, CEM, DCR and LICEM models, the training process involved utilizing an AdamW optimizer (Loshchilov & Hutter, 2017). The $\lambda_y$ coefficient (2) was set to 0.5 to emphasize concept learning over task loss while $\lambda_w = 1 \times 10^{-6}$ and $\lambda_b = 10^{-6}$. Moreover, a scheduler was implemented with a gamma of 0.1 and a step size of 10 epochs throughout the training period of 100 epochs. After every hidden layer we have used a ReLU activation function. Here are further insights into the methodologies' architectures, with the number of output neurons indicated within brackets.

- E2E: layer 1 (100), layer 2 (number of classes);

- CEM: concept embedding size of 768, layer 1 (10), layer 2 (number of classes);

- CBMs, concept prediction: layer 1 (10), layer 2 (number of concepts);

    - LL, task prediciton: layer (number of classes);
    - MLP, task prediction: layer 1 ($3 \cdot$ number of concepts), layer 2 (number of classes).

- DCR: the temperature parameter is set to $0.1$.

The text's embedding size varies depending on the chosen backbone. When employing BERT, it remains at 768, whereas adopting the LLMs approach (Jiang et al., 2023b) it increases to 4096. For Dtree and XGBoost, we employed the default hyperparameter settings. The DTree model was implemented using the sklearn library, while the XGBoost model was implemented using the xgboost library[4]. We conducted five experiments for each methodology. The training time for the different experiments averages around 10 minutes using the setup specified in Section 4.1.

The CEBaB dataset (Abraham et al., 2022) does not necessitate any splitting procedure as it inherently offers training, validation, and test sets. In the training set, modifications include counterfactual examples, while both the validation and test sets exclusively contain original reviews. For the remaining datasets, we partitioned the data into training, validation, and test sets using stratified sampling based on the task labels. The proportions allocated are 0.7 for training, 0.1 for validation, and 0.2 for testing. Each experiment was conducted with a different seed.

A.4    ENCODER COMPARISON

This section presents all the results obtained using a fine-tuned BERT backbone as the encoder $h(x)$. In the remainder of the paper, we consistently reported results when utilizing *Mixtral 8x7B* (Jiang et al., 2024) as the backbone model. In this section, we instead provide the performance of all models in terms of task accuracy (see Table 3) and of concept macro-averaged F1 score (refer to Table 4) when employing BERT as the backbone (Devlin et al., 2018), which is an encoder-only model.

Both tables show that there is no great difference with respect to Tables 1, 6, with BERT providing slightly lower performance on Multiemo-It and on the Drug dataset. This result shows that the proposed approach can be applied also to other architectures. We chose to employ Mixtral in the remainder of the paper since it can be also effectively used to provide concept annotations, therefore having a single model for both encoding the sample and predicting the concept scores.

---

[4]The xgboost library we used can be found at `https://github.com/dmlc/xgboost`.

Table 3: This table presents the performance in terms of task accuracy (%) of different models utilizing BERT as backbone. We report in **bold** the best result among the same type of models (e.g., supervised, interpretable ones) considering models equally best if their standard deviations overlap. We use ✓to indicate models requiring concept supervision (C. Sup.) or having an interpretable task predictor (T. Inter.). We highlight in light gray the models we propose in this work. We do not report supervised model results for depression ($-$) since it does not provide concept annotations.

| Type | Method | C. Sup. | T. Inter. | CEBaB | Multiemo-It | Drug | Depression |
|---|---|---|---|---|---|---|---|
| E2E | MLP | ✗ | ✗ | $\textbf{90.68}_{\pm 0.47}$ | $\textbf{75.67}_{\pm 0.47}$ | $\textbf{59.33}_{\pm 0.56}$ | $\textbf{97.80}_{\pm 0.23}$ |
| SUP. | CBM+MLP | ✓ | ✗ | $78.01_{\pm 6.51}$ | $54.10_{\pm 4.51}$ | $36.67_{\pm 6.24}$ | $-$ |
| | CBM+XG | ✓ | ✗ | $80.00_{\pm 0.34}$ | $69.02_{\pm 0.64}$ | $51.00_{\pm 0.28}$ | $-$ |
| | CEM | ✓ | ✗ | $\textbf{90.67}_{\pm 0.47}$ | $\textbf{77.00}_{\pm 0.82}$ | $\textbf{58.33}_{\pm 1.70}$ | $-$ |
| | CBM+LL | ✓ | ✓ | $61.00_{\pm 12.02}$ | $49.67_{\pm 5.46}$ | $34.33_{\pm 7.38}$ | $-$ |
| | CBM+DT | ✓ | ✓ | $75.67_{\pm 0.47}$ | $65.02_{\pm 0.34}$ | $46.23_{\pm 0.78}$ | $-$ |
| | DCR | ✓ | ✓ | $86.55_{\pm 0.58}$ | $74.01_{\pm 0.24}$ | $59.75_{\pm 0.45}$ | $-$ |
| | LICEM (ours) | ✓ | ✓ | $\textbf{87.89}_{\pm 0.38}$ | $\textbf{75.31}_{\pm 0.15}$ | $\textbf{60.14}_{\pm 0.44}$ | $-$ |
| GEN. | CBM+MLP | ✗ | ✗ | $73.93_{\pm 5.67}$ | $44.19_{\pm 2.07}$ | $35.16_{\pm 4.3}$ | $83.20_{\pm 2.18}$ |
| | CBM+XG | ✗ | ✗ | $83.29_{\pm 0.43}$ | $69.85_{\pm 1.55}$ | $34.94_{\pm 0.91}$ | $87.00_{\pm 1.01}$ |
| | CEM | ✗ | ✗ | $\textbf{85.88}_{\pm 0.95}$ | $\textbf{73.15}_{\pm 0.67}$ | $\textbf{56.95}_{\pm 0.36}$ | $\textbf{96.12}_{\pm 0.50}$ |
| | CBM+LL | ✗ | ✓ | $58.81_{\pm 7.16}$ | $58.35_{\pm 1.59}$ | $36.84_{\pm 11.52}$ | $51.48_{\pm 2.16}$ |
| | CBM+DT | ✗ | ✓ | $79.28_{\pm 0.52}$ | $62.61_{\pm 2.08}$ | $34.17_{\pm 0.11}$ | $80.55_{\pm 0.03}$ |
| | DCR | ✗ | ✓ | $\textbf{85.63}_{\pm 0.81}$ | $70.02_{\pm 2.70}$ | $57.46_{\pm 0.02}$ | $95.98_{\pm 0.27}$ |
| | LICEM (ours) | ✗ | ✓ | $\textbf{86.22}_{\pm 0.66}$ | $\textbf{74.45}_{\pm 0.57}$ | $\textbf{60.23}_{\pm 0.58}$ | $\textbf{96.87}_{\pm 0.20}$ |

Table 4: This table presents the performance in terms of concept prediction of the models that utilize BERT as backbone. Concept prediction (%) of the compared models for datasets equipped with concept annotations is measured using the macro-averaged F1 score. We report in **bold** the best result among the same type of models (e.g., supervised, interpretable ones) considering models equally best if their standard deviations overlap. We highlight in light gray the models we propose in this work. The methods using the self-generative have the same macro-averaged F1 score, therefore we use $-$ to represent all methods.

| Type | Method | CEBaB | Multiemo-It | Drug |
|---|---|---|---|---|
| E2E | MLP | $\textbf{79.92}_{\pm 1.77}$ | $\textbf{63.25}_{\pm 1.09}$ | $\textbf{79.01}_{\pm 2.9}$ |
| SUP. | CBM+MLP | $75.17_{\pm 3.11}$ | $\textbf{64.08}_{\pm 1.22}$ | $74.26_{\pm 0.9}$ |
| | CEM | $\textbf{79.97}_{\pm 1.29}$ | $\textbf{64.42}_{\pm 1.21}$ | $\textbf{77.32}_{\pm 1.2}$ |
| | CBM+XG | $\textbf{79.92}_{\pm 1.77}$ | $63.25_{\pm 1.09}$ | $\textbf{79.01}_{\pm 0.9}$ |
| | CBM+LL | $74.25_{\pm 4.55}$ | $62.08_{\pm 0.88}$ | $73.11_{\pm 1.7}$ |
| | CBM+DT | $79.92_{\pm 1.77}$ | $63.25_{\pm 1.09}$ | $79.01_{\pm 2.9}$ |
| | DCR | $82.06_{\pm 0.40}$ | $64.29_{\pm 0.42}$ | $80.10_{\pm 0.2}$ |
| | LICEM (ours) | $\textbf{82.93}_{\pm 0.13}$ | $\textbf{65.61}_{\pm 0.69}$ | $\textbf{81.59}_{\pm 0.42}$ |
| GEN. | CBM+MLP | $75.05_{\pm 8.31}$ | $49.59_{\pm 10.01}$ | $43.58_{\pm 14.99}$ |
| | CEM | $\textbf{81.08}_{\pm 0.44}$ | $\textbf{58.30}_{\pm 1.79}$ | $\textbf{80.99}_{\pm 0.42}$ |
| | CBM+XG | $79.24_{\pm 1.21}$ | $\textbf{60.79}_{\pm 0.71}$ | $64.72_{\pm 0.45}$ |
| | CBM+LL | $\textbf{78.75}_{\pm 0.59}$ | $\textbf{61.72}_{\pm 0.24}$ | $66.72_{\pm 19.48}$ |
| | CBM+DT | $\textbf{79.24}_{\pm 1.21}$ | $\textbf{60.79}_{\pm 0.70}$ | $64.72_{\pm 0.45}$ |
| | DCR | $\textbf{80.25}_{\pm 1.02}$ | $59.11_{\pm 0.84}$ | $\textbf{81.47}_{\pm 0.49}$ |
| | LICEM (ours) | $77.79_{\pm 2.49}$ | $58.87_{\pm 0.66}$ | $\textbf{81.18}_{\pm 0.33}$ |
| SELF GEN. | $-$ | $\textbf{84.08}_{\pm 0.00}$ | $\textbf{64.27}_{\pm 0.00}$ | $\textbf{83.00}_{\pm 0.00}$ |

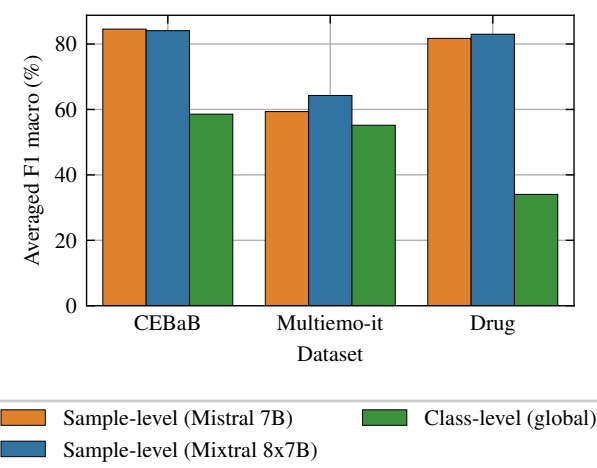

Figure 10: Comparison among concept annotation methods where the annotation quality is measured in terms of macro-averaged F1 score. On average, Mixtral 8x7B yields the best results.

### A.5    LLM-BASED CONCEPT ANNOTATION VS CLASS-LEVEL ANNOTATION

This section presents a comparison between the usage of two different LLMs, *Mistral 7B* (Jiang et al., 2023a) and *Mixtral 8x7B* (Jiang et al., 2024), as concept annotators. In Figure 10 we report the results in terms of macro-averaged F1 score (as concept classes are highly imbalanced) on the three datasets for which human concept annotation is available. We also report, as a baseline, a global (class-level) annotation strategy, providing to all samples belonging to a given class the same concept annotation. In this case, we label the positive class with positive concepts and negated negative concepts (e.g. for all samples of the class *Good Drug* we use 'Efficient' and 'Not Side Effects'). We can observe that between the two LLMs there is not a significant difference in performance, with Mixtral 8x7B providing on average slightly better results. Comparing against the baseline, instead, we can observe that there is a great improvement in CEBaB and in the Drug dataset, while in Multiemo-It the improvement is more modest.

### A.6    TASK ACCURACY AND CONCEPTS PREDICTION PERFORMANCE

In this section we report the task accuracy and the concepts prediction performance results for all the different experiments conducted, generative approach included when using Mixtral 8x7B as a backbone. As shown in Table 5, LICEM outperforms the other task interpretable models, reaching the highest task accuracy for the CEBaB dataset using the generative approach.

We also report the averaged F1 macro to measure the concepts prediction performance of all models when provided with all the available concept annotations. The results shown in Figure 3 are here confirmed. We again see that Self-supervised strategy is a very good approach since without human effort it provides better concept macro-averaged F1 score in CEBaB and Drug. Only on Multiemo-It the performance are significantly lower. This result may be due to the fact that the latter dataset is in Italian while the other datasets are in English, a language for which the LLMs have certainly seen more training samples.

### A.7    SURVEY CHARACTERIZATION

In this section, we provide further details regarding the conducted survey. A total of 46 participants with varying levels of experience in machine learning, from complete beginners to experts, were recruited (see Figure 11). The gender distribution was nearly balanced, with $40\%$ identifying as female and $60\%$ as male. The majority of participants, $91.3\%$, were within the $20-40$ age range, while only $8.7\%$ were aged over $40$.

The survey was structured in the following manner:

Table 5: Task accuracy (%) of the compared models. We report in **bold** the best result among the same type of models (e.g., supervised, interpretable ones) considering models equally best if their standard deviations overlap. We use ✓to indicate models requiring concept supervision (C. Sup.) or having a task-interpretable predictor (T. Inter.). We highlight in light gray the models we propose in this work. The Generative and the Self-generative approaches extend the scalability of concept-based models to datasets without concept annotations, where supervised models cannot be applied ($-$).

| Type | Method | C. Sup. | T. Inter. | CEBaB | Multiemo-It | Drug | Depression |
|------|--------|---------|-----------|-------|-------------|------|------------|
| E2E | Mixtral–MLP | ✗ | ✗ | **88.80** $\pm$ 0.75 | 80.01 $\pm$ 0.63 | **63.66** $\pm$ 1.20 | **97.18** $\pm$ 0.03 |
| | Mixtral–Zero-shot | ✗ | ✗ | 86.80 $\pm$ 0.31 | 80.06 $\pm$ 0.66 | 60.81 $\pm$ 0.28 | 73.77 $\pm$ 0.23 |
| | Mixtral–Few-shot | ✗ | ✗ | 84.79 $\pm$ 0.42 | **84.17** $\pm$ 0.67 | 62.16 $\pm$ 0.27 | 76.38 $\pm$ 0.08 |
| SUP. | CBM+MLP | ✓ | ✗ | 78.41 $\pm$ 9.30 | 45.43 $\pm$ 8.20 | 45.42 $\pm$ 4.90 | $-$ |
| | CBM+XG | ✓ | ✗ | 83.01 $\pm$ 0.10 | 69.01 $\pm$ 0.02 | 55.00 $\pm$ 0.13 | $-$ |
| | CEM | ✓ | ✗ | **89.60** $\pm$ 0.49 | **83.33** $\pm$ 0.47 | **66.81** $\pm$ 0.40 | $-$ |
| | CBM+LL | ✓ | ✓ | 71.43 $\pm$ 9.71 | 42.67 $\pm$ 7.01 | 34.60 $\pm$ 10.10 | $-$ |
| | CBM+DT | ✓ | ✓ | 77.20 $\pm$ 0.40 | 65.00 $\pm$ 0.02 | 47.20 $\pm$ 0.40 | $-$ |
| | DCR | ✓ | ✓ | 88.05 $\pm$ 0.53 | 82.01 $\pm$ 0.71 | 65.40 $\pm$ 0.80 | $-$ |
| | LICEM (ours) | ✓ | ✓ | **89.89** $\pm$ 0.77 | **83.47** $\pm$ 0.49 | **66.80** $\pm$ 0.29 | $-$ |
| GEN. | CEM | ✗ | ✗ | **89.97** $\pm$ 0.66 | **82.41** $\pm$ 0.11 | 63.80 $\pm$ 0.38 | **97.06** $\pm$ 0.11 |
| | CBM | ✗ | ✓ | 62.07 $\pm$ 0.22 | 68.66 $\pm$ 4.20 | 33.14 $\pm$ 2.10 | 50.25 $\pm$ 0.39 |
| | DCR | ✗ | ✓ | 88.97 $\pm$ 0.18 | 80.82 $\pm$ 0.54 | 63.74 $\pm$ 1.16 | 95.35 $\pm$ 0.21 |
| | LICEM (ours) | ✗ | ✓ | **90.64** $\pm$ 0.38 | 81.85 $\pm$ 0.71 | **66.15** $\pm$ 0.44 | 96.50 $\pm$ 0.18 |
| SELF GEN. (OURS) | Self-CBM+MLP | ✗ | ✗ | 82.71 $\pm$ 0.01 | 75.42 $\pm$ 4.42 | 47.59 $\pm$ 0.33 | 82.31 $\pm$ 0.04 |
| | Self-CBM+XG | ✗ | ✗ | 82.70 $\pm$ <0.01 | 79.09 $\pm$ <0.01 | 53.28 $\pm$ <0.01 | 82.28 $\pm$ <0.01 |
| | Self-CEM | ✗ | ✗ | **89.14** $\pm$ 0.38 | **84.06** $\pm$ 0.09 | 65.20 $\pm$ 0.73 | **97.16** $\pm$ 0.08 |
| | Self-CBM+LL | ✗ | ✓ | 82.71 $\pm$ 1.23 | 77.15 $\pm$ 0.96 | 47.35 $\pm$ 0.29 | 82.12 $\pm$ 0.15 |
| | Self-CBM+DT | ✗ | ✓ | 83.95 $\pm$ <0.01 | 78.44 $\pm$ <0.01 | 53.28 $\pm$ <0.01 | 82.28 $\pm$ <0.01 |
| | Self-DCR | ✗ | ✓ | 87.72 $\pm$ 0.66 | 83.47 $\pm$ 0.43 | 63.29 $\pm$ 0.36 | **97.11** $\pm$ 0.03 |
| | Self-LICEM | ✗ | ✓ | **89.56** $\pm$ 0.29 | **84.49** $\pm$ 0.25 | 65.89 $\pm$ 0.39 | **97.23** $\pm$ 0.21 |

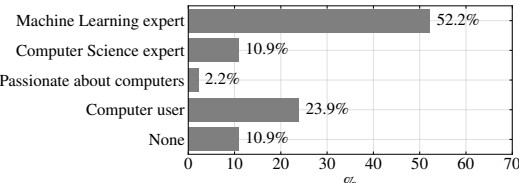

Figure 11: Distribution of users by expertise level.

- **Introduction to Explanations**: We provided an introduction to the various types of explanations, ensuring that participants had sufficient background information to understand and interpret these explanations.

- **Questionnaire**: Participants were asked a total of 7 questions for each of the three datasets that contained english text: CEBaB, Drug, and Depression. The questions were divided as follows:

  - The first 3 questions asked participants to select their preferred explanation for a given text. Examples of these questions can be found in Figure 6, 7, 8.

  - The remaining 4 questions asked participants to predict the label of the text based on a provided explanation, with two questions pertaining to DCR and two to LICEM. Examples of these questions are presented in Figure 9, 12, 13.

For both types of questions, we randomly selected samples from the three datasets (CEBaB, Drug, and Depression) where both models (LICEM and DCR) made the correct predictions.

Table 6: This table presents the performance in terms of concept prediction of the models that utilize Mixtral 8x7B as backbone. Concept prediction (%) of the compared models for datasets equipped with concept annotations is measured using the macro-averaged F1 score. We report in **bold** the best result among the same type of models (e.g., supervised, interpretable ones) considering models equally best if their standard deviations overlap. Self-supervised methods are reported with the same concept accuracy with zero standard deviation, since the concept predictions are provided by an LLM with temperature set to zero. The methods using the self-generative have the same macro-averaged F1 score, therefore we use $-$ to represent all methods.

| Type | Method | CEBaB | Multiemo-It | Drug |
|---|---|---|---|---|
| E2E | MLP | $75.92_{\pm 0.77}$ | $74.25_{\pm 1.02}$ | $78.50_{\pm 0.23}$ |
| SUP. | CBM+MLP | $65.17_{\pm 2.35}$ | $61.75_{\pm 1.02}$ | $65.33_{\pm 2.46}$ |
| | CEM | $78.83_{\pm 0.85}$ | $77.12_{\pm 1.38}$ | $80.79_{\pm 0.47}$ |
| | CBM+XG | $75.92_{\pm 0.77}$ | $74.25_{\pm 1.02}$ | $78.50_{\pm 0.23}$ |
| | CBM+LL | $64.25_{\pm 2.56}$ | $59.12_{\pm 2.13}$ | $64.83_{\pm 1.20}$ |
| | CBM+DT | $75.92_{\pm 0.77}$ | $74.25_{\pm 1.02}$ | $78.50_{\pm 0.23}$ |
| | DCR | $78.45_{\pm 1.92}$ | $75.67_{\pm 1.43}$ | $79.96_{\pm 0.43}$ |
| | LICEM (ours) | $75.45_{\pm 0.93}$ | $76.36_{\pm 0.39}$ | $80.83_{\pm 0.36}$ |
| GEN. | CBM+MLP | $71.87_{\pm 0.14}$ | $52.60_{\pm 14.32}$ | $55.68_{\pm 19.84}$ |
| | CEM | $74.70_{\pm 0.98}$ | $63.61_{\pm 0.44}$ | $79.45_{\pm 0.41}$ |
| | CBM+XG | $75.02_{\pm 0.57}$ | $61.69_{\pm 0.44}$ | $79.15_{\pm 0.30}$ |
| | CBM+LL | $72.15_{\pm 0.59}$ | $63.72_{\pm 0.84}$ | $66.72_{\pm 19.48}$ |
| | CBM+DT | $75.02_{\pm 0.57}$ | $61.69_{\pm 0.44}$ | $79.04_{\pm 0.30}$ |
| | DCR | $75.62_{\pm 2.59}$ | $62.79_{\pm 0.44}$ | $79.04_{\pm 0.33}$ |
| | LICEM (ours) | $74.44_{\pm 0.25}$ | $63.75_{\pm 0.36}$ | $79.05_{\pm 0.58}$ |
| SELF GEN. | $-$ | $84.08_{\pm 0.00}$ | $64.27_{\pm 0.00}$ | $83.00_{\pm 0.00}$ |

According to the explanation, select the correct label. *

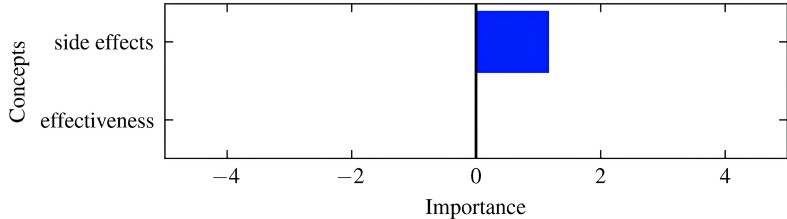

Figure 12: Example of label prediction given LICEM explanation, Drug dataset.

## A.8 CONCEPT INTERVENTIONS

As introduced in Section 4.4, LICEM is sensible to concept interventions. This characteristic is very important since it implies that a human can interact with the model, providing counterfactual predictions when prompted with different concept predictions. In Figure 14, 15, 16 we simulate this situation by correcting mispredicted concepts with the correct concept predictions and check whether the task prediction has been also modified. More in details, we report the improvement in task accuracy when increasing the probability to correct the concepts, demonstrating LICEM's

According to the explanation, select the correct label.                    *

Explanation: Deprecation ∧ ¬ Loss_of_Interest ∧ ¬ Sleep_Disturbances ∧ ¬ Appetite_Changes

○ Depressed comment

○ Not depressed comment

Figure 13: Example of label prediction given DCR explanation, CEBaB dataset.

responsiveness and significant performance improvement. A similar behaviour can also be observed for CBMs, even though they were starting from a lower task accuracy and a higher increase could have also been expected. For comparison, we also report the E2E model with a flat line, since it does not offer this possibility. As noted in (Espinosa Zarlenga et al., 2022), CEMs (which are not task interpretable) may not respond well to concept interventions, especially without conducting them during training. Thus, we trained all CEM-based models with a 0.5 intervention probability during the forward pass.

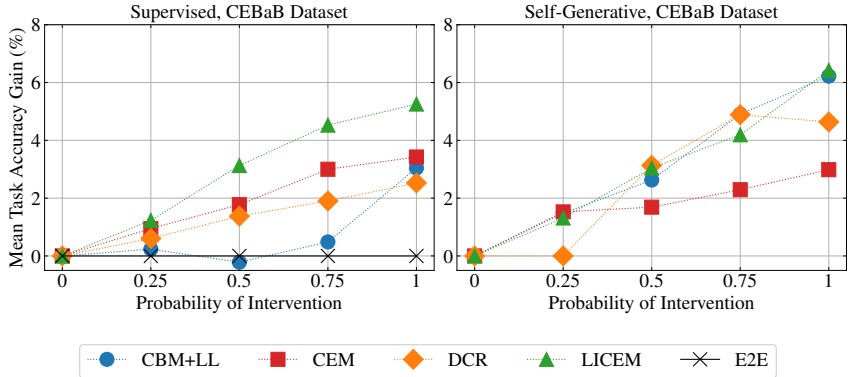

Figure 14: Concept interventions on the CEBaB dataset for (left) supervised approaches and (right) self-supervised ones.

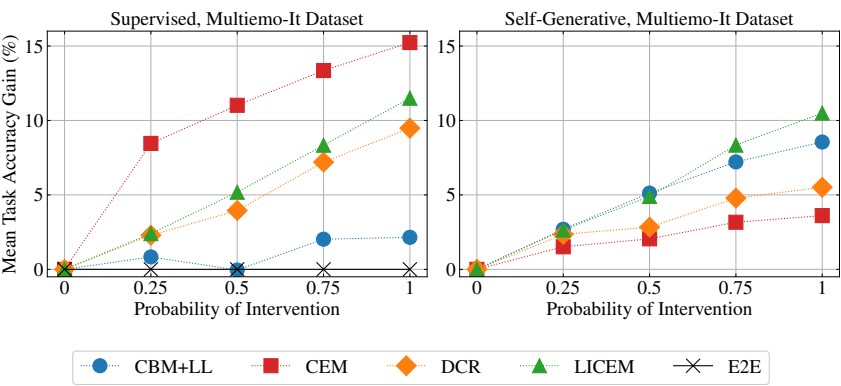

Figure 15: Concept interventions on the Multiemo-it dataset for (left) supervised approaches and (right) self-supervised ones.

### A.9 CAUSAL CONCEPT EFFECT (CACE)

As anticipated in Section 4.4, Concept-based models predictions must be causally influenced by the predicted concepts. We assess concept-based models' responsiveness to *do-interventions* using the Causal Concept Effect (CaCE) (Goyal et al., 2019), which measures the impact of input modifications

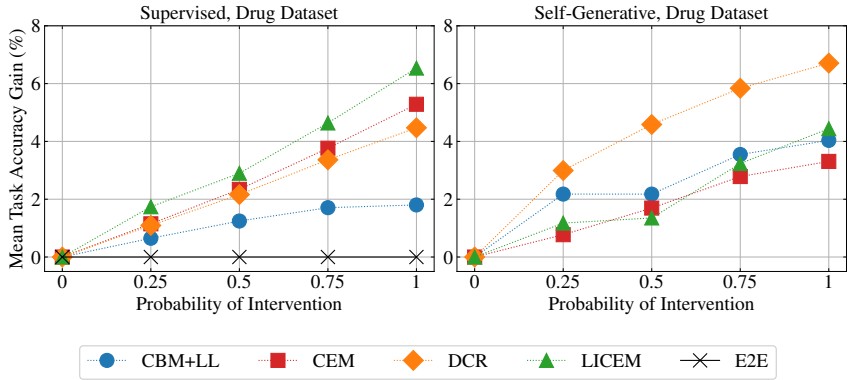

Figure 16: Concept interventions on the Drug dataset for (left) supervised approaches and (right) self-supervised ones.

Table 7: Causal Concept Effect (CaCE) for different methods. A high (absolute) value implies a strong responsiveness of a model to modifications to a certain concept.

| | Concept | CBM+LL | CEM | DCR | LICEM | SELF-LICEM |
|---|---|---|---|---|---|---|
| **CeBAB** | Good Food | -0.02 ± 0.01 | 0.29 ± 0.03 | 0.33 ± 0.04 | 0.62 ± 0.02 | 0.63 ± 0.01 |
| | Good Amb. | 0.01 ± 0.05 | 0.08 ± 0.01 | 0.02 ± 0.01 | 0.18 ± 0.03 | 0.20 ± 0.04 |
| | Good Service | 0.01 ± 0.04 | 0.13 ± 0.01 | 0.20 ± 0.08 | 0.37 ± 0.01 | 0.35 ± 0.02 |
| | Good Noise | -0.01 ± 0.10 | -0.05 ± 0.01 | -0.02 ± 0.01 | 0.15 ± 0.02 | 0.15 ± 0.03 |
| **Multiemo** | Joy | 0.04 ± 0.06 | 0.18 ± 0.01 | 0.16 ± 0.07 | 0.28 ± 0.01 | 0.27 ± 0.01 |
| | Trust | 0.02 ± 0.10 | 0.60 ± 0.04 | 0.47 ± 0.15 | 0.62 ± 0.03 | 0.63 ± 0.01 |
| | Sadness | -0.04 ± 0.05 | -0.06 ± 0.01 | -0.04 ± 0.02 | -0.04 ± 0.01 | -0.10 ± 0.02 |
| | Surprise | -0.01 ± 0.06 | 0.03 ± 0.01 | 0.06 ± 0.05 | -0.02 ± 0.01 | 0.01 ± 0.01 |
| **Drug** | Effectiveness | 0.02 ± 0.10 | 0.43 ± 0.02 | 0.28 ± 0.02 | 0.45 ± 0.04 | 0.46 ± 0.02 |
| | Side Effects | -0.07 ± 0.14 | -0.52 ± 0.01 | -0.25 ± 0.02 | -0.55 ± 0.06 | -0.55 ± 0.03 |

on model predictions. Higher absolute CaCE values indicate stronger conditioning on relevant concepts. Tables 7 shows that both supervised and self-supervised LICEM have higher CaCE values compared to CBM, CEM and DCR, suggesting stronger reliance on predicted concepts. This result is positive since all concepts considered in this work are relevant for the task at hand. We leave for future work the exploration of tasks where there are confounding concepts and checking whether LICEM is capable to not consider them.

