# OpenReview forum: "Linearly Interpretable Concept Embedding Model for Text Classification"
_ICLR.cc/2025/Conference — ICLR 2025 Conference Withdrawn Submission_

### Official Review · Reviewer_noRf · 2024-10-19

**Soundness:** 2
**Presentation:** 2
**Contribution:** 2
**Rating:** 3
**Confidence:** 2

**Summary:**

The paper introduces a model, Linearly-Interpretable Concept Embedding Model (LICEM), in order to enhance the interpretability in text classification. It uses a self-supervised approach to generate human-interpretable concepts, eliminating the need for extensive labeled concept data. The experimental results prove that this model matches black-box models performance, is interpretable and can be trained without concept supervision.

**Strengths:**

The proposed method achieves better performance than previous methods.

**Weaknesses:**

This work uses an LLM as a task predictor in Section 3.1, and produces self-generation in Section 3.2. However, LLMs' generations may contain hallucination, so the generations of LLMs should not be trusted. This work aims at increasing the interpretability, so the groundtruth is very important. It is not convincing when using one blackbox to interpret another blackbox.

The contribution seems to be limited. The performance is enhanced by using a LLM-based CEM to replace the original text encoder. However, this increasement is not surprising due to LLMs' strong ability.

**Questions:**

How can you ensure that the self-generated concepts are correct?

---

### Official Review · Reviewer_4NFr · 2024-11-01

**Soundness:** 3
**Presentation:** 3
**Contribution:** 2
**Rating:** 6
**Confidence:** 4

**Summary:**

This paper introduces the Linearly Interpretable Concept Embedding Model (LICEM), which enhances the interpretability of neural networks by incorporating embeddings and concepts components generated by LLMs. The original LICEM trains a prediction layer to minimize both the task loss and the concept prediction loss. In contrast, the advanced self-LICEM, leveraging LLMs for concept prediction, only needs to minimize the task loss. The proposed method is good at both prediction interpretation and classification performance.

**Strengths:**

1. The paper is well written and organized.
2. The experiments are comprehensive.
3. The proposed method is good at both prediction interpretation and classification performance.

**Weaknesses:**

1. Although the final prediction is somewhat interpretable, intermediate steps like the concept distribution provided by LLMs remain uninterpretable, making the entire system still lack explainability.
2. The framework is limited to classification tasks.
3. The paper includes a user study that may expose participants to potentially harmful content, such as datasets related to depression or drugs. Therefore, an ethics statement is needed to address these concerns.

**Questions:**

For self-LICEM, since the LLM provides almost everything, including embeddings and concept distributions, why do we still need an MLP head for predictions? Why not utilize the LLM itself to complete the entire framework, including prediction, like using Chain-of-thought, prompt engineer or SFT the model with the dataset? Introducing an additional MLP head seems to diminish the utility of the LLM's original LM head.

**Details Of Ethics Concerns:**

The paper includes a user study that may expose participants to potentially harmful content, such as datasets related to depression or drugs. Therefore, an ethics statement is needed to address these concerns.

---

### Official Review · Reviewer_SfK7 · 2024-11-04

**Soundness:** 2
**Presentation:** 3
**Contribution:** 2
**Rating:** 5
**Confidence:** 4

**Summary:**

The paper introduces a Linearly Interpretable Concept Embedding Model (LICEM) aimed at improving interpretability in text classification while maintaining high accuracy. Existing Concept-Bottleneck Models (CBMs) which often require manual concept annotations and face limitations in interpretability with non-linear predictors, LICEM uses a linear equation to classify text to provide better interpretability, and use a LLM to generate concepts. Experimental results suggest LICEM achieves comparable performance with black-box models on several text classification datasets while improving explainability.

**Strengths:**

1. The paper improves over existing CEM by providing a more human-understandable linear combination of concept embeddings.
2. LICEM achieves comparable results over black-box models on 4 text classification datasets, without requiring pre-annotated concepts.

**Weaknesses:**

1. LICEM was tested on only four text classification tasks. As LICEM does not require manual concept annotation, evaluating it on more diverse datasets from general domains, such as the GLUE benchmark or 20 Newsgroups, would help verify the claim that LICEM can match the performance of black-box models for short text classification while offering better interpretability.
2. While LICEM’s use of linearly interpretable concept embeddings is valuable, this approach has been explored in previous works, such as in Interpreting Embedding Spaces by Conceptualization.

**Questions:**

1. The paper lacks a detailed analysis of LICEM’s efficiency. What is the computational cost associated with concept self-generation? what is the dimensionality of the concept embeddings, and how does it affect overall performance?
2. What is the difference between LICEM and CEM with a linear layer?
3. Could the authors provide details on the number of concepts generated for each task when using generative and self-generative approaches? This would help clarify scalability and feasibility across different tasks.
4. How does LICEM handle previously unseen but important concepts that may appear at test time? Addressing this would strengthen LICEM’s applicability to real-world settings.

---

### Official Review · Reviewer_k6w8 · 2024-11-05

**Soundness:** 3
**Presentation:** 3
**Contribution:** 3
**Rating:** 5
**Confidence:** 4

**Summary:**

This paper introduces the Linearly Interpretable Concept Embedding Model (LICEM) for text classification, designed to improve interpretability without sacrificing classification accuracy. Traditional explainability methods in large language models (LLMs) often rely on post-hoc approaches like attention and gradient analysis, which have been found to provide limited insights. Although Concept-Bottleneck Models (CBMs) have been proposed for interpretable predictions, they suffer from limitations in accuracy, task interpretability, and the requirement of extensive annotations. LICEM addresses these issues by offering a linearly interpretable model that makes predictions based on concept embeddings, enabling high accuracy and interpretability without extensive concept labeling. Experimental results and a user study demonstrate that LICEM outperforms existing interpretable models and achieves similar or better performance than black-box models.

**Strengths:**

1. LICEM advances interpretability in LLMs by using concept embeddings within a linear framework. This design choice enhances the clarity of the model’s decision-making process, enabling users to understand which concepts influence predictions directly.

2. LICEM achieves high accuracy levels that match or exceed those of existing black-box models. By bridging the gap between accuracy and interpretability, LICEM offers a valuable improvement over traditional concept-bottleneck models, which often struggle with reduced classification performance.

3. The inclusion of a user study that evaluates the plausibility and usefulness of LICEM's explanations strengthens the claim of improved interpretability.

**Weaknesses:**

1. In this work, the authors mainly consider text classification. However, although the provided interpretations could help researchers understand the reasoning process during text classification, more complex tasks like natural language inference could be more suitable for evaluation. This is because generation tasks are more prevalent and crucial in the evaluation of LLMs.

2. The use of instance-level linear equations for interpretability might introduce scalability challenges with large datasets or complex models. An analysis of LICEM’s scalability and computational efficiency would provide clearer insights into its practical feasibility for large-scale applications.

3. This work cannot be applied to black-box LLMs, which are recently more powerful than smaller LMs that are white-box. The authors should consider an alternative strategy that is suitable for also black-box LLMs.

**Questions:**

See the weaknesses.

---

### Official Review · Reviewer_47Ep · 2024-11-05

**Soundness:** 2
**Presentation:** 3
**Contribution:** 2
**Rating:** 5
**Confidence:** 5

**Summary:**

The paper presents LICEM model for text classification that addresses the interpretability challenges in LLMs, which often lack transparency in decision making process. Unlike existing Concept-Bottleneck Models (CBMs), LICEM provides task-relevant explanations based on human-understandable concepts without sacrificing classification accuracy. This is achieved by utilizing a linear equation for predictions over concept embeddings, improving interpretability and enabling plausible explanations. A self-generative variant Self-LICEM is also proposed to removes the need for manual concept annotations by leveraging LLMs to predict concepts directly.

**Strengths:**

- LICEM offers a transparent, linearly interpretable structure, allowing users to understand how different concepts contribute to predictions.

- The model maintains accuracy comparable to black-box models while enhancing interpretability.

- A  Self supervised LICEM variant reduces dependency on annotated concepts, enabling deployment in cases where annotations are unavailable.

**Weaknesses:**

- LICEM is evaluated mainly on text classification; I'm wonderingif it could generlize to other domains. the assumption that concepts linearly relate to the output may not hold in complex tasks.

- It is not clear if self-LICEM  will inherit biases or hallocination from the used LLMs and how to deal with that. As it relies on prompt-responses from LLMs for concept prediction, this can introduce variability to the specific prompts used.

- the concept embeddings may capture task-related signals from text classification. This could be seen as concept leakage too, which may reducing interpretability.

**Questions:**

- Compared to CBMs, how the model deal with high dimensional concept spaces?
- What is the motivation behind fixing the temperature of  Self generated LICEM as 0?

---

### Note · Authors · 2024-11-25

**Comment:**

We appreciate the reviewers' valuable feedback. We will enhance our work based on their suggestions and plan to resubmit it to a future conference. Thank you once again!

**Withdrawal Confirmation:**

I have read and agree with the venue's withdrawal policy on behalf of myself and my co-authors.